# Polypharmacy Is Associated with Lower Memory Function in African American Older Adults

**DOI:** 10.3390/brainsci10010049

**Published:** 2020-01-16

**Authors:** Shervin Assari, Cheryl Wisseh, Mohammed Saqib, Mohsen Bazargan

**Affiliations:** 1Department of Family Medicine, Charles R Drew University of Medicine and Science, Los Angeles, CA 90095, USA; mobazarg@cdrewu.edu; 2Department of Pharmacy Practice, West Coast University School of Pharmacy, Los Angeles, CA 91606, USA; cWisseh@westcoastuniversity.edu; 3Health Behavior & Health Education, University of Michigan, Ann Arbor, MI 48109, USA; saqimoha@umich.edu; 4Department of Family Medicine, University of California Los Angeles (UCLA), Los Angeles, CA 90095, USA

**Keywords:** race, ethnicity, African American, aging, elderly, older adults, medication use, polypharmacy, memory, cognitive function, Black

## Abstract

Although previous research has linked polypharmacy to lower cognitive function in the general population, we know little about this association among economically challenged African American (AA) older adults. This study explored the link between polypharmacy and memory function among AA older adults. This community-based study recruited 399 AA older adults who were 65+ years old and living in economically disadvantaged areas of South Los Angeles. Polypharmacy (taking 5+ medications) was the independent variable, memory function was the outcome variable (continuous variable), and gender, age, living arrangement, socioeconomic status (educational attainment and financial strain), health behaviors (current smoking and any binge drinking), and multimorbidity (number of chronic diseases) were the covariates. Linear regression was used for data analyses. Polypharmacy was associated with lower scores on memory function, above and beyond covariates. Among AA older adults, polypharmacy may be linked to worse cognitive function. Future research should test the mechanisms by which polypharmacy is associated with lower levels of cognitive decline. There is a need for screening for memory problems in AA older adults who are exposed to polypharmacy.

## 1. Introduction

Polypharmacy (P), commonly defined as taking 5+ medications a day [1], is common in low socioeconomic status (SES) contexts, racial/ethnic minority people, and advanced ages (i.e., older adults) [2,3]. Although polypharmacy can be defined in various ways, and each definition may seem arbitrary, a systemic review of definitions of polypharmacy showed that using 5+ a day is the most commonly accepted definition of polypharmacy in medical research [1]. 

Polypharmacy increases the risk of a wide range of undesirable health outcomes and events such as adverse drug events and drug–drug interactions [4,5]. Polypharmacy is associated with reduced medication adherence [6,7,8,9] and an increase in potentially inappropriate medication (PIM) use [10]. Polypharmacy is also associated with undesirable clinical and health outcomes such as falls [11], hospitalization [12], emergency department visits [12,13], cognitive decline [14,15,16], and premature mortality [17,18,19]. Given the negative health consequences of inappropriate polypharmacy, researchers have shown interest in understanding factors that are associated with polypharmacy. 

Although polypharmacy is more common in older adults [20,21], very limited information is available on factors that correlate with polypharmacy in African American (AA) older adults [2,3,22]. The main reason polypharmacy is more common in older adults is that multimorbidity, which is required for polypharmacy [20,21], is more common in older adults. Polypharmacy is particularly common in AAs, who often develop chronic diseases (CDs) at an earlier age than Whites [23,24]. To fill the existing gap in the literature, there is a need to study the correlation of polypharmacy in low-income AA older adults with multimorbidity. 

Although polypharmacy is associated with cognitive decline and memory loss [14,15,25,26], minimal information exists on how polypharmacy contributes to memory loss in AA older adults. While some cross-sectional and longitudinal studies have suggested that polypharmacy may be a risk factor for cognitive changes in the general population [14,15,25,26], we still need to explore the same association in economically challenged AA older adults. If polypharmacy is also responsible for some of the cognitive decline of AA older adults with multimorbidity, then prevention of inappropriate polypharmacy becomes a core strategy to prevent the cognitive/memory decline of this population. 

Memory loss is a major problem in older adults [27,28], including AA older adults [27,29,30,31]. Memory loss, which is different from dementia [32], reduces the quality of life [28], impairs disease management [33], predicts mortality [34], and predicts a wide range of undesirable health outcomes [28,35]. Memory loss is also a source of economic burden to families, health care systems, and society [36,37]. Although inappropriate medication may operate as a risk factor for memory loss, this association is not commonly investigated in AA older adults. 

The main aim of this study was to investigate the association between polypharmacy (taking 5+ medications) [1] and memory loss in AA older adults in economically challenged areas of Los Angeles. We tested this association net of age, gender, SES, health behaviors, and health (CDs), all of which may confound the association between polypharmacy and a specific domain of cognitive function, namely memory function. 

## 2. Materials and Methods

The current cross-sectional study, which was conducted in South Los Angeles, CA, USA, was composed of two sections: (1) a self-reported survey (interview) that collected data on demographics, SES, physical health, and mental health, and (2) a comprehensive evaluation of medications taken.

### 2.1. Health Survey 

The first part of this study was composed of a survey that took about 60 min to answer and was made up of the multi-item responses to health questionnaires on various topics such as chronic disease, substance use, mental health, pain, sleep, exercise, diet, quality of life, etc. 

### 2.2. Comprehensive Evaluation of Medications

The second part of this study included a comprehensive evaluation of all medications that were being used by the participants. This went beyond taking a simple medication history as all medications and their dosages were recorded by a physician or a nurse. This part of the data collection was not self-reported, and medications were provided by the patients. These included both PRN (As Needed) and scheduled medications and included both over the counter (OTC) and prescribed (RX). Given the nature of the data collection, we defined our medication data collection as comprehensive. Data were gathered between 2015 and 2018 [10,38,39,40,41,42,43].

### 2.3. Interviews and Language

The participants needed to answer the survey items themselves. No help from a translator or a caregiver was acceptable. In addition, participants needed to conduct a full interview in English. This was decided because our participants were all AAs, defined as Blacks who were born and raised in the US. Thus, no other language was provided.

The ethical aspects of this study received approval by the Charles R. Drew University (CDU) Institutional Review Board (IRB number = 14-12-2450-05). All the participants provided signed informed consent at the time of enrollment in the study. Participants also received some financial compensation. As mentioned above, the data were collected in two sections: (1) a structured face–to–face interview for evaluation of health, and (2) a comprehensive investigation of all medications that were being taken by the participant. 

Our study used consecutive, non-random sampling to enroll participants. Participants were enrolled if they were (1) AA, (2) older adults defined as 65 years or older, (3) living in economically challenged areas in South Los Angeles, (4) could conduct a full health interview in English. Exclusion criteria included (1) institutionalized individuals, (2) enrollment in other clinical trials (to prevent any arbitrary change in the study variables due to the clinical trials). This approach increased the external validity of our results to the general population. The current analysis included 399 AA older adults (age ≥ 65 years).

### 2.4. Sampling in an Economically Challenged Area and Sampling Frame

Participants were recruited from 16 predominantly AA churches, 11 senior housing apartment units, and some low-income public housing projects located in Service Planning Area (SPA) 6, LA County. Participation of the individuals was facilitated and encouraged by church leaders and housing apartment managers. Given that SPA6 is among the most economically disadvantaged areas, we were able to recruit community-dwelling members that were historically underserved and low income. The vast majority of older adults in SPA6 are AA (49%). From about 10.3 million individuals who live in LA County, 1.3 million are older adults. As LA County is the most populous county in the US, and given its large size (4300 square miles), LA County is divided into eight Service Planning Areas (SPAs). These SPAs have a more homogenous economic and demographic composition. These distinct regions facilitate surveillance and provision of public health and clinical services in a more tailored manner. Approximately 28–36% of SPA 6 households are below the federal poverty level and are uninsured. In SPA 6, about 60% of adults have income levels less than 200% of the federal poverty line (FPL), which is 1.5 times more than LA County overall. As we drew participants from SPA 6, participants were economically challenged. Across various SPAs, SPA 6 has the highest percentage of low-income AA and Hispanic people. Overall, SPA 6 is the most economically disadvantaged area of LA County. SPA 6 has recently experienced a rapid increase in the percentage of homeless AA individuals. Although all of LA County is experiencing an increase in homelessness, this is a more severe challenge in the poorest areas where AA individuals are more likely to lose their homes due to an inability to pay mortgage, rent, or property tax. Given poverty is concentrated in SPA6, homelessness has increased in this section of LA County [39]. 

### 2.5. Measures

The study variables included age, gender, SES, multimorbidity, polypharmacy, and memory function. 

*Demographic covariates*. Gender, age, and living arrangement were the demographic variables. Gender (male 1, female 0) and living arrangement (living alone 1, living with others 0) were binary variables. Age was a continuous variable.

*Socioeconomic status*. The two SES indicators in this study were educational attainment and financial strain. Educational attainment (years of schooling) was a continuous variable. To measure education attainment, we counted all years of schooling, including not only primary but also higher education (college years). Financial strain was measured using three items that measure whether participants lack enough money for what is essential to meet primary needs, namely (1) food, (2) clothes, and (3) utility bills. Responses for each item were on a 5-point Likert scale (never = 1, always = 5). We built a sum score with a range between 3 and 15. A high score reflected more financial strain (lower SES). Cronbach’s alpha of this study was 0.923 [44].

*Current Cigarette Smoking*. Participants’ smoking habits (of cigarettes) was measured using the following item “How would you describe your cigarette smoking habits?” Responses included never-smokers, ex-smokers (individuals who use to smoke but successfully quitted), and current smokers. We calculated a dichotomous variable: current smoker 1, never/past smoker 0.

*Alcohol Binge Drinking*. Participants were asked if they drink alcohol. They were also asked about the frequency of their drinking. Individuals who answered positively to the question of whether they drink alcohol were asked: “How often did you have six or more drinks?” Responses ranged from 0 to 6 which were for never, a few times a year, every few months, monthly, every few weeks, weekly, and daily. Due to having a very low number of individuals who had high levels of binge drinking in the sample, we categorized binge drinking as a dichotomous variable: 0 no alcohol binge drinking vs. 1 alcohol binge drinking. This was also operationalized as dichotomous variables [45,46]. 

*Multimorbidity (Number of Chronic Disease (CDs))*. Participants’ multimorbidity was measured as the number of CDs the participant has. Participants were asked which of the following (if any) CDs they had: hypertension or high blood pressure, heart diseases including coronary artery disease, diabetes mellitus, hypercholesterolemia and lipid disorders, cancer and tumors, asthma and chronic obstructive pulmonary disease (COPD), rheumatoid arthritis, osteoarthritis, thyroid disorders, and gastrointestinal disease. The presence of CDs was counted regardless of the level of activity of the illness. Participants reported whether any physician or health care provider had ever informed them that they have the above chronic diseases. Multimorbidity was treated as a continuous rather than a categorical variable, defined as the total number of CDs, with a potential range between 0 and 10. 

*Polypharmacy*. Polypharmacy was defined as taking 5+ medications per day [1]. This was determined based on a comprehensive evaluation of medications.

*Outcome*. Our main dependent variable was memory function, which was measured using the Meta Memory Score of daily forgetfulness (MMQ-Ability). The MMQ-Ability uses 20 items, with a score ranging from 1 to 80. A higher score was indicative of higher memory function while lower scores were indicative of memory dysfunction (Cronbach’s α 0.909) [47,48].

To perform data analysis, we used SPSS 23.0 (IBM, Armonk, New York, NY, USA). To describe the participants, we reported mean, standard deviation (SD), and frequency (n, %), depending on the variable type. For bivariate analysis, we used the Spearman correlation test. This test helped us test for any possible collinearity between the study variables. As Table 2 shows, there was no collinearity between any variables (Pearson correlation coefficient [*r* value] between polypharmacy and multimorbidity was 0.35). For multivariable analysis, a linear regression model was used. In our multivariable models, polypharmacy was the predictor variable, memory function was the dependent variable, and demographics, SES, and health were the covariates. We used the Enter selection so variables remained in the model even when they were not statistically significant. Beta (regression coefficient), standard errors (SE), 95% CI, and polypharmacy values were reported. *p* values equal to or smaller than 0.05 were considered significant. 

## 3. Results

Table 1 describes the sample characteristics. The mean age of the participants was 73 (SD = 7) years. The sample included 399 AA older adults (age ≥ 65 years) which were either men (*n* = 141; 35.3%) or women (*n* = 258; 64.7%). Overall, 75.4% of the participants were implementing polypharmacy (5+ medications per day). On average, participants were taking 7.6 (SD = 4) medications, with a range from 2 to 22. Memory function showed a range from 1 to 80 with a mean score of 52.11 (SD = 12.50). 

### 3.1. Unadjusted Bivariate Correlations 

Table 2 shows a correlation matrix between study variables. There was a negative correlation between polypharmacy and memory function. Gender and number of CDs (multimorbidity) also showed a correlation with memory function. Age, educational attainment, financial strain, living arrangement, binge drinking, and smoking did not show a significant correlation with memory function (Table 2).

### 3.2. Adjusted Multivariable Associations

Table 3 shows the results of multivariable analysis, with polypharmacy as a categorical independent variable, memory function as a continuous outcome, and all other variables as covariates. In this model, polypharmacy was associated with lower memory function, after adjusting for all covariates. After all other variables were adjusted, individuals with polypharmacy reported scores lower by 3.06 units on their memory function compared with individuals without polypharmacy. In addition to polypharmacy, a higher number of CDs (multimorbidity) was associated with a lower memory function, and male gender was associated with higher memory function. Each additional CD was associated with 1 additional unit decline in memory function. Compared to females, men reported 3 units higher in memory function (Table 3).

## 4. Discussion

This study revealed an association between polypharmacy and lower memory function in economically disadvantaged AA older adults, above and beyond SES and health confounders that included age, gender, SES, and health (multimorbidity). 

The finding that low-income AA older adults with polypharmacy are at an increased risk of memory loss is in line with the literature that suggests polypharmacy may be one of the contributing factors to cognitive and memory decline [49]. In a study on people above 65 years old, authors found that polypharmacy results in the decline of cognitive performance in about 50% of the patients who were tested in rapid mental status check [49]. In a study conducted in Austria, polypharmacy was not linked to cognitive decline; however, hyper-polypharmacy (>10 medications) was positively associated with low cognitive performance [16]. In Tokyo, Japan, polypharmacy predicted cognitive impairment in a community sample of older adults who were residing in urban areas [50]. In a study of 498 dementia-free participants around 64 years old, polypharmacy was significantly more common in individuals with cognitive impairment. This association was independent of confounders such as comorbidities [15]. In the SHELTER study, which stands for “Services and Health for Elderly in Long TERm care”, conducted in Europe and Israel, polypharmacy was predictive of worsened cognitive function but not functional decline [51].

Research suggests that clinicians and practitioners who work with economically disadvantaged AA older adults should know about the link between polypharmacy and memory impairment in their population. Although more research is needed, inappropriate polypharmacy should be screened and prevented in all low SES AA older adults, particularly those with memory loss. Thus, the results of this study can help physicians/prescribers to reduce their likelihood of prescribing inappropriate polypharmacy. 

A side observation was that more than half of the sample population were living alone, considering the AA community is associated with a collective instead of an individualistic culture. We also observed a mean of 12.8 years of education, which suggests that our sample was a well-educated group. Participants were still living in low SES areas, suggesting that education has smaller than expected effects on changing AA life conditions [52,53,54,55,56]. 

Our study is with some limitations. First, this was a cross-sectional study. Such a design rules out any causal inferences. Thus, our result on the link between polypharmacy and memory loss should be interpreted as an association rather than causation. Other limitations included low sample size and non-random sampling. Another limitation was reliance on self-reported data on multimorbidity. As mentioned before, we solely relied on self-reporting to measure CDs. Although some biases are possible, research has shown the high validity of self-reported CDs [57]. Future research, however, should try to verify the self-reported measure of multimorbidity with medical claims and charts. In addition, we only measured the quantity, not the quality, of polypharmacy. Without knowing which cases of polypharmacy are inappropriate or appropriate, it is difficult to apply the results to health promotion and prevention of polypharmacy and memory loss. Future research should explore how inappropriate polypharmacy reduces cognitive function and health overall through drug interactions. In addition, not all potential confounders were included in our analysis. For example, we did not have data on somatization and symptoms. We also did not have data on frailty. As our sample was not random, and as we used a convenient sample, we cannot generalize our findings to the community of AA older adults in the US. As a result of these limitations, future research should replicate the finding reported here in nationally representative data. This study also only included AAs. A strength of this study was evaluating medications and not relying on self-reporting of medications. However, given the age group of the participants and some cognitive decline of the sample, some of the study variables, such as number of CDs, may be less reliable than other variables, such as polypharmacy. Other ethnic groups could provide additional insight into how these effects differ across groups [58]. The strength of the link between polypharmacy and cognitive function may differ across various ethnic groups. Given these limitations, we need more studies. The results reported here should be regarded as preliminary. The results extend the literature from White to low-income AA older adults.

### 4.1. Future Research

There are high-risk and low-risk medications for memory and cognition decline. High-risk medications include benzodiazepines and anticholinergic agents. Future research should test the specific types of medications that are linked to more severe memory impairment. Similarly, there is a need to study domains of cognition other than memory function. In our study, we only had the number of medications, thus further analysis should also focus on the type of medications. Future research should also study other confounders within and beyond medications and chronic diseases.

### 4.2. Conclusions

In summary, polypharmacy is associated with memory loss in economically disadvantaged AA older adults. Interventions that wish to address cognitive decline in particular and memory loss, in particular among AA older adults, may benefit from addressing medication-related challenges such as polypharmacy. There is a need for interventions that test the effects of programs on cognitive function in AA older adults. Future research is needed to understand whether drug–drug interaction is responsible for the link between polypharmacy and various aspects of cognitive decline beyond memory loss, and if we can undo this link for AA older adults.

## Figures and Tables

**Table 1 brainsci-10-00049-t001:** Descriptive characteristics.

	*n*	%
Gender		
Female	258	64.7
Male	141	35.3
Living Alone		
No	197	49.4
Yes	202	50.6
Binge Drinking		
No	357	89.5
Yes	42	10.5
Smoking		
No	346	86.7
Yes	53	13.3
Polypharmacy (P)		
No	98	24.6
Yes	301	75.4
Number of Pharmacy		
1.00	286	71.7
2.00	86	21.6
3.00	15	3.8
4.00	8	2.0
6.00	1	0.3
7.00	1	0.3
Number of Health Care Providers		
1.00	124	31.1
2.00	122	30.6
3.00	91	22.8
4.00	37	9.3
5.00	15	3.8
6.00	5	1.3
7.00	2	0.5
8.00	3	0.8
	**Mean**	**SD**
Age	73.52	6.98
Educational attainment (1–16)	12.75	2.34
Financial strain	6.71	3.69
Multimorbidity (Number of CDs: 0–10)	3.45	1.80
Number of medications (2–22)	7.63	3.99
Memory function (1–80)	52.11	12.50

CDs: Chronic diseases.

**Table 2 brainsci-10-00049-t002:** Bivariate correlates of memory function.

	1	2	3	4	5	6	7	8	9	10
1 Gender (Male) (0/1)	1.00	−0.04	−0.11 *	−0.02	−0.08	0.16 **	0.27 **	−0.10 *	−0.19 **	0.13 *
2 Age (Years) (65–95)		1.00	−0.20 **	−0.13 **	0.07	−0.13 **	−0.27 **	0.00	0.07	0.04
3 Education attainment (1–16)			1.00	−0.08	−0.01	−0.01	−0.02	−0.07	0.00	0.03
4 Financial strain (5–25)				1.00	0.05	0.12 *	0.15 **	0.12 *	−0.04	−0.08
5 Living arrangement (Living alone)					1.00	0.06	−0.04	−0.02	−0.05	0.02
6 Any binge drinking (0/1)						1.00	0.32 **	−0.07	−0.05	0.04
7 Current smoking (0/1)							1.00	−0.10 *	−0.15 **	−0.03
8 Number of CDs (Multimorbidity: 0–10)								1.00	0.35 **	−0.20 **
9 Polypharmacy (2–22)									1.00	−0.16 **
10 Memory function (1–80)										1.00

* *p* < 0.05, ** *p* < 0.01.

**Table 3 brainsci-10-00049-t003:** Linear regression with memory function as the outcome.

	B (Regression Coefficient)	Std. Error	95% CI	*p*
Gender (Male)	3.07	1.37	0.38–5.76	0.025
Age (Years)	0.07	0.10	−0.12–0.25	0.494
Education attainment	0.21	0.28	−0.34–0.76	0.457
Financial strain	−0.17	0.17	−0.51–0.17	0.339
Living arrangement (living alone)	0.35	1.24	−2.09–2.79	0.778
Drinking (any binge)	1.68	2.13	−2.51–5.87	0.430
Smoking (current)	−3.24	2.04	−7.26–0.78	0.114
Number of CDs (multimorbidity)	−1.02	0.37	−1.76–−0.29	0.006
Polypharmacy	−3.06	1.56	−6.12–0.00	0.050
Constant	50.62	8.97	32.98–68.25	0.000

Dependent variable: Memory function.

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
