# Peer review of "Polypharmacy Is Associated with Lower Memory Function in African American Older Adults"

_brainsci, 2020, doi:10.3390/brainsci10010049_

Round 1

Reviewer 1 Report

Thank you for the opportunity to review an original article reporting polypharmacy in African American older adults with a low socioeconomic status.  The study attempted to address an important knowledge gap in a vulnerable group and the article contribute to our limited understanding between polypharmacy and memory loss.  Please see comments below. 

Introduction: This section was presented clearly.  It would be benefit from a succinct rationale why memory loss is a problem in the population of interest and expand on inappropriate medication as a risk factor.  Also need to be clear with the terminology, memory loss is not the same as dementia.  Need careful consideration. 

Specific comments for this section:

Page 1, line 37-39: This last sentence is redundant at it form as the purpose of it is unclear as suggested in the following sentence that polypharmacy is a key risk.  Expand what are the other risk factors besides multimorbidity. Page 1, line 42-43: repetition to sentences in the previous para.

Materials and methods: This section could be improved by clearer description of the study sample, i.e. What % of the sampling frame were AA aged 65 plus? What are the recruitment strategies? is the % of the total study sample?

A main statistical question is what approach was undertaken to deal with collinearity between polypharmacy and multi-morbidity in regression model.  A statement from the authors to clarify this is required.

Specific comments:

Page 2, line 72: Please clarify why enrolment on other clinical trials is an exclusion criterion. Page 2, line 88: It is unclear if the Cronbach’s alpha value is a results from the current study (should be in the result) or from previous study. Page 2, line 92: ‘ex-smoker’: this option is ambiguous open to different interpretation. Page 3, line 102: More description on the method of comprehensive medication evaluation is require. Page 3, line 105: Cronbach’s alpha value, Is this from the current study? Confusing. Page 3, line 107: More description is require for the multiple regression technique, e.g. which step-wise approach?

Results: This section report interesting results.  One that particular caught my attention was more than half of the sample live alone considering AA is associated as a collective culture instead of individualistic culture.  From an international perspective, a mean of 12.8 years education suggests this is a rather well educated group. 

Specific comments:

Table 1; please report number of medications in this sample. Is the median number of medication similar to 5?  Please include descriptive stats (mean (SD) or median (IQR)) score for the MMQ-Ability Table 3: Is this the fully adjusted model? Please comment how you deal with co-linearity between polypharmacy and multimorbidity. Looking at this table, there is no statistical association between polypharmacy and memory function, p=0.05

Discussion: validity of these responses considering literacy level and memory loss. The main discussions presented here focus on cognitive function and the study outcome is memory loss, which is a subset of cognitive function.  Need to be cautious that memory loss is not interchangeable with cognitive function. 

Specific comments:

Page 5, line 156: Reference "Agreement between self-reports and medical records of cardiovascular disease in octogenarians." J Clin Epidemiol 66: 1135-1143. Line 166: Be cautious with the interpretation, the study sample include AA with low SES, not all AA. Line 167: What leads you to this hypothesis? Some references are require. A suggestion of ref "Self-rated health, health related behaviours and medical conditions of Māori and non-Māori in advanced age: LiLACS NZ." N Z Med J 127(1397): 13-29. Line 172: Intervention proposed is applicable for the physicians/prescribers.

Author Response

R1

Thank you for the opportunity to review an original article reporting polypharmacy in African American older adults with a low socioeconomic status.  The study attempted to address an important knowledge gap in a vulnerable group and the article contribute to our limited understanding between polypharmacy and memory loss. Please see comments below. 

Thank you. The comments you provided have made a paper much better and we appreciate it a lot!

Introduction: This section was presented clearly.  It would be benefit from a succinct rationale why memory loss is a problem in the population of interest and expand on inappropriate medication as a risk factor.  Also need to be clear with the terminology, memory loss is not the same as dementia.  Need careful consideration. 

Thank you. We made some modifications to our introduction! We cited papers showing memory function is a problem in this population and inappropriate medication as a risk factor. We are clear that memory loss is not the same as dementia.

Specific comments for this section:

Page 1, line 37-39: This last sentence is redundant at it form as the purpose of it is unclear as suggested in the following sentence that polypharmacy is a key risk.  Expand what are the other risk factors besides multimorbidity. Page 1, line 42-43: repetition to sentences in the previous para.

We tried to improve the structure and flow of the introduction and eliminate the redundancy in this part of the paper. These have impacted the lines 37-39 and 42-43.

Materials and methods: This section could be improved by clearer description of the study sample, i.e. What % of the sampling frame were AA aged 65 plus? What are the recruitment strategies? is the % of the total study sample?

We have added a more clear description of our sample, our recruitment strategy, our SPA6, sampling frame, etc.

A main statistical question is what approach was undertaken to deal with collinearity between polypharmacy and multi-morbidity in regression model.  A statement from the authors to clarify this is required.

There was no collinearity between polypharmacy and multi-morbidity in the study, as shown in the Table 2.

Specific comments:

Page 2, line 72: Please clarify why enrolment on other clinical trials is an exclusion criterion.

We explained why we have made this decision. This was made because we wanted to avoid any additional confounder in our study.

Page 2, line 88: It is unclear if the Cronbach’s alpha value is a results from the current study (should be in the result) or from previous study.

We are clear that Cronbach’s alpha value is from the current study.

Page 2, line 92: ‘ex-smoker’: this option is ambiguous open to different interpretation.

We added the description of what we need with ‘ex-smoker’.

Page 3, line 102: More description on the method of comprehensive medication evaluation is require.

We added a few sentences explaining what we did when we say “comprehensive medication evaluation”.

Page 3, line 105: Cronbach’s alpha value, Is this from the current study? Confusing.

From this study. We mentioned this in the text.

Page 3, line 107: More description is required for the multiple regression technique, e.g. which step-wise approach?

We used the Enter approach which keeps the variables in the model regardless of their significance. That means we have only got 1 output set which we have presented.  

Results: This section report interesting results.  One that particular caught my attention was more than half of the sample live alone considering AA is associated as a collective culture instead of individualistic culture.  From an international perspective, a mean of 12.8 years education suggests this is a rather well educated group. 

We added a part to the discussion regarding these points that most people were educated and living alone.

Specific comments:

Table 1; please report number of medications in this sample. Is the median number of medication similar to 5?  Please include descriptive stats (mean (SD) or median (IQR)) score for the MMQ-Ability Table 3: Is this the fully adjusted model? Please comment how you deal with co-linearity between polypharmacy and multimorbidity. Looking at this table, there is no statistical association between polypharmacy and memory function, p=0.05

We added average of the MMQ-Ability (memory function) as well as number of medications to the table 1 as well as the text of the results section.

Discussion: validity of these responses considering literacy level and memory loss. The main discussions presented here focus on cognitive function and the study outcome is memory loss, which is a subset of cognitive function.  Need to be cautious that memory loss is not interchangeable with cognitive function. 

We added to our discussion that given the individuals had some degrees of memory loss, some of the variables may not be very valid.

Specific comments:

Page 5, line 156: Reference "Agreement between self-reports and medical records of cardiovascular disease in octogenarians." J Clin Epidemiol 66: 1135-1143.

We cited as mentioned.

Line 166: Be cautious with the interpretation, the study sample include AA with low SES, not all AA.

Here and there, we have added “economically disadvantaged” or low income or low SES to describe our sample in the discussion so people are repeatedly reminded that the study sample include AA with low SES, not all AA.

Line 167: What leads you to this hypothesis? Some references are require. A suggestion of ref "Self-rated health, health related behaviours and medical conditions of Māori and non-Māori in advanced age: LiLACS NZ." N Z Med J 127(1397): 13-29.

We cited as mentioned.

Line 172: Intervention proposed is applicable for the physicians/prescribers.

We mentioned that the intervention proposed is applicable for the physicians/prescribers.

Reviewer 2 Report

The authors should be commended on their work to describe the link between polypharmacy and cognitive impairment in the economically challenged AA population.  However, more information is needed to fully convey what was done in this study to the reader.

Intro:

Line 31: Polypharmacy can be defined in various ways.  Please provide some explanation of the arbitrariness of using 5 as a cutpoint.

Line 35: In the geriatric literature, PIMs (potentially inappropriate medication) is used instead of IUM.  May consider changing to align with standard terms.  Since many PIMs are linked to cognitive impairment, how did you limit this exposure if your population ?

Materials and Methods:

Line 70: what is meant by economically challenged area?  Is there a standard definition to economically challenged?

Line 71: Did the participant need to answer the survey or could a caregiver help them?  Also, why was English needed to conduct the interviews?  One would think that in this area, more languages would be provided.

Line 72: Why were people excluded in they were in other clinical trials?

Line 73: What is an SPA?  Is it a clinic?  Is it a governmental organization?  Please explain.

Line 77: Why is there a rapid increase in the percent of homeless people in SPA 6?

Line 84: Education attainment, did this include only high school or did it include college years as well?

Line 85: What were the 3 items that measure financial strain?  Food, rent/mortgage, clothes and bills would count for 4 items, so please clarify how this was measured.

Line 89: How was binge drinking assessed?  Please add in more information.

Line 94: What is a CD?  Is this chronic disease?  Where the disease states verified?  Without verification, there could be a lot of bias in what the patients report.  Please explain.

Line 100: Was each condition tallied or was each category tallied?  For instance, what if a patient had two active cancers, is this 1 or 2?  Please clarify.

Line 102: what is meant by a comprehensive evaluation of medications?  Was a medication history taken by a pharmacist?  Were meds self-reported?  Were these PRN or scheduled meds?  OTC or RX or both?  Much more information is needed to assess how the medications were collected.  Without an explanation, the internal validity of this study is poor.  Did you calculate an anticholinergic burden to asses the risk of meds on cognition?

Line 103: How did you classify someone as having cognitive impairment?  What cutpoint did you use?  Or was this entered in as a continuous variable?  What is considered normal?  More inform in needed to let the reader know how you determined your primary outcome measure.

Table 1:

What is the average number of meds?

Table 2: what do the * represent?

Table 3: what is the significance of B, please explain. How does one interpret these results?

Discussion:

There are high risk meds for cognition including benzo's and anticholinergic agents.  Were these linked to cognitive impairment?  Just having a number of meds is ok, but what further analysis did you do to limit other confounders within the meds themselves?

Author Response

The authors should be commended on their work to describe the link between polypharmacy and cognitive impairment in the economically challenged AA population.  However, more information is needed to fully convey what was done in this study to the reader.

Thanks.

Intro:

Line 31: Polypharmacy can be defined in various ways.  Please provide some explanation of the arbitrariness of using 5 as a cutpoint.

We added this explanation to our intro.

Line 35: In the geriatric literature, PIMs (potentially inappropriate medication) is used instead of IUM.  May consider changing to align with standard terms.  Since many PIMs are linked to cognitive impairment, how did you limit this exposure if your population ?

Now we use PIMs.

Materials and Methods:

Line 70: what is meant by economically challenged area?  Is there a standard definition to economically challenged?

We have added a full paragraph on this issue to our methods.

Line 71: Did the participant need to answer the survey or could a caregiver help them?  Also, why was English needed to conduct the interviews?  One would think that in this area, more languages would be provided.

No caregivers. The participants answered. This is mentioned in the paper.

Line 72: Why were people excluded in they were in other clinical trials?

To reduce additional bias. This is explained now.

Line 73: What is an SPA?  Is it a clinic?  Is it a governmental organization?  Please explain.

One paragraph explains this now. Please see the Methods section. 

Line 77: Why is there a rapid increase in the percent of homeless people in SPA 6?

Explained at the same section of SPA6.

Line 84: Education attainment, did this include only high school or did it include college years as well?

Yes. Added to the methods section. 

Line 85: What were the 3 items that measure financial strain?  Food, rent/mortgage, clothes and bills would count for 4 items, so please clarify how this was measured.

Made clear now. 

Line 89: How was binge drinking assessed?  Please add in more information.

Definition and measurement added.

Line 94: What is a CD?  Is this chronic disease?  Where the disease states verified?  Without verification, there could be a lot of bias in what the patients report.  Please explain.

Added as a limitations. and citations are added too.

Line 100: Was each condition tallied or was each category tallied?  For instance, what if a patient had two active cancers, is this 1 or 2?  Please clarify.

CDs were counted regardless of severity and chronicity. 

Line 102: what is meant by a comprehensive evaluation of medications?  Was a medication history taken by a pharmacist?  Were meds self-reported?  Were these PRN or scheduled meds?  OTC or RX or both?  Much more information is needed to assess how the medications were collected.  Without an explanation, the internal validity of this study is poor.  Did you calculate an anticholinergic burden to asses the risk of meds on cognition?

We added a section on this comprehensive evaluation of medications. No self-report.  PRN and scheduled meds + OTC and RX were assessed. 

Line 103: How did you classify someone as having cognitive impairment?  What cutpoint did you use?  Or was this entered in as a continuous variable?  What is considered normal?  More inform in needed to let the reader know how you determined your primary outcome measure.

Added. We have not use a dichotomous variable. We have used linear regression and this is explained. memory is used as a continuous measure. 

Table 1:

What is the average number of meds?

Added.

Table 2: what do the * represent?

Added to the table.

Table 3: what is the significance of B, please explain. How does one interpret these results?

Explained how it should be interpreted. 

Discussion:

There are high risk meds for cognition including benzo's and anticholinergic agents.  Were these linked to cognitive impairment?  Just having a number of meds is ok, but what further analysis did you do to limit other confounders within the meds themselves?

We explained there are low risk and high risk meds regarding cognitive / memory impairment.